# Dietary Pediocin Supplementation Restores Intestinal Barrier Function and Microbiota Balance in *Salmonella*-Infected Specific-Pathogen-Free Chickens

**DOI:** 10.3390/microorganisms14010018

**Published:** 2025-12-20

**Authors:** Chenxin Zhou, Hui Liu, Bowen Yang, Zefeng Zhang, Mingrong Zhang, Siyue Zhang, Zhihua Feng, Dongyan Zhang

**Affiliations:** 1College of Animal Science and Technology, Hebei Agricultural University, Baoding 071001, China; zhoucx0222@163.com (C.Z.); yangbowen@hebau.edu.cn (B.Y.); 18733215360@163.com (Z.Z.); 2Institute of Animal Science and Veterinary Medicine, Beijing Academy of Agriculture and Forestry Sciences, Beijing 100097, China; liuh1860@sina.com; 3College of Plant Protection, Hebei Agricultural University, Baoding 071001, China; 19981647734@163.com; 4College of Bioscience and Resources Environment, Beijing University of Agriculture, Beijing 102206, China; zhangsiyue0330@163.com

**Keywords:** chickens, pediocin, *S. pullorum*, intestinal barrier, gut microbiota

## Abstract

In this study, the effects of pediocin (PP) on intestinal barrier function, renal injury, and immune regulation were evaluated in *Salmonella pullorum*-infected chickens. Forty-five 7-day-old specific-pathogen-free (SPF) chickens were randomly assigned to three groups: control (CON), *S. pullorum* infection (SP), and *S. pullorum* infection + PP treatment (SPA). The results showed that *S. pullorum* infection significantly elevated (*p* < 0.05) the renal (CREA, UREA), hepatic (ALT, AST), immunological (IgG, IgM), and inflammatory (TNF-α, IL-6, SAA, CRP) parameters, as well as the expression of *trefoil factor 3*, *Toll-like receptor 2*, *TNF-α*, *IL-1β*, and *IL-6*. In contrast, the jejunal villus height and the villus-to-crypt ratio, and the expression of intestinal tight junction proteins (*occludin*, *claudin-1*, and *Zonula occludens-1*), *mucin-2*, and *transforming growth factor-β1* were significantly decreased in both the SP and SPA groups. In the SP group, the parameter alterations observed at 6 DPI compared to the CON group persisted until 12 DPI. In contrast, in the SPA group, these parameters returned to levels comparable to those of the CON group after 6 days of PP treatment. Moreover, *S. pullorum* infection markedly reduced the α-diversity of the gut microbiota, and this reduction could be partially restored following PP treatment. At the phylum level, *S. pullorum* infection significantly reduced the relative abundances of Proteobacteria and Verrucomicrobia. PP treatment increased the abundances of Firmicutes and Actinobacteria, while also restoring the abundances of Proteobacteria and Verrucomicrobia to some extent. At the genus level, PP treatment significantly increased the abundance of *Faecalibacterium* and *Lactobacillus*. Additionally, *Faecalibacterium* and *Butyricicoccus* were significantly more abundant in the SPA group. Thus, PP could alleviate *S. pullorum* infection induced intestinal barrier damage, reduce immune stress responses, and exert a protective effect by modulating the composition of the intestinal microbiota of chickens.

## 1. Introduction

*Salmonella*, a common foodborne pathogen, poses a direct threat to animal health, causes substantial economic losses, and is a potential risk to human food safety [1]. *Salmonella* Pullorum is primarily parasitic in poultry hosts and can cause pullorum disease, a septicemic condition with an extremely high mortality rate in chicks and turkey poults. Transmission mainly occurs vertically (through eggs) but can also spread horizontally within poultry environments. In laying hens infected with *S. pullorum*, the most notable effects include decreased egg production, reduced egg weight, and poorer eggshell quality (such as thinner shells and higher breakage rates) [2,3], leading to economic losses in poultry farming. Therefore, effective health management and disease control are paramount in poultry farming. Recent epidemiological studies in China have confirmed *S. pullorum* as the most prevalent serotype in large-scale poultry farms [4,5]. Antibiotics are widely used in livestock farming to shorten disease duration and limit pathogen transmission [6]. However, this strategy has several drawbacks. Prolonged antibiotic use not only promotes the development of antimicrobial resistance in *Salmonella* and other enteric bacteria but also disrupts the composition of the gut host microbiota, leading to long-term adverse effects on animal health [7,8,9]. The European Union prohibited antibiotic use as growth promoters in 2006, and China introduced a similar ban on 1 July 2020. Since the prohibition, the treatment of intestinal diseases poses a major challenge in the poultry industry. Identifying alternative therapeutic approaches has become a central focus in current research to overcome these challenges. Among potential alternatives, bacteriocins such as nisin [10], which are produced by *Lactococcus lactis*, have demonstrated efficacy in inhibiting the growth of *Salmonella*. Nevertheless, their mechanisms of action and practical applications in the prevention and control of *Salmonella* infections remain insufficiently studied. Therefore, further investigation and development of novel antimicrobial agents, particularly bacteriocin-based alternatives, are important in mitigating the threat of *Salmonella*, reducing antibiotic dependence, and safeguarding both animal and human health.

Bacteriocins are ribosomally synthesized small peptides or proteins that exhibit potent antimicrobial activity, low host cytotoxicity, and a minimal likelihood of inducing drug resistance. These natural and safe bio-antimicrobial agents have been widely used in the fields of medicine [11], food processing [12], and animal husbandry [13]. Bacteriocins modulate the intestinal microbiota by inhibiting pathogenic bacteria while promoting the proliferation of beneficial bacteria, thereby maintaining intestinal health, enhancing immune responses, and alleviating physiological stress [14,15]. Mechanistically, most bacteriocins inactivate target cells by disrupting cell membranes and forming pores [16]. The bacteriocin pediocin from *Pediococcus acidilactici* is broad-spectrum and thermally stable [17] with proven efficacy in poultry. In broilers, dietary pediocin A alleviates Clostridium perfringens-induced growth impairment and tends to lower ADG (average daily gain) loss [18]. In chickens, supplementation of *Pediococcus acidilactici* (PA) enhances early production performance and eggshell quality [19]. Additionally, *Pediococcus pentosaceus* DYNDL69M8, which contains a pediocin-like operon, has been shown to increase beneficial gut bacteria and raise acetic acid concentration in mice, as demonstrated by Cui et al. [20]. However, bacteriocins, including nisin and pediocin, their precise mechanism of action on intestinal barrier function in vitro has not yet been completely elucidated.

Previously, an engineered strain of *Saccharomyces cerevisiae* producing pediocin was successfully constructed. This strain exhibited excellent antibacterial properties in vitro and had a particularly strong antagonistic effect against *Salmonella*. However, it remains unclear as to whether bacteriocins can enhance gut health in a *Salmonella*-challenge model of chickens. Therefore, the efficacy of *Pediococcus pentosaceus*-derived bacteriocins in controlling *Salmonella* infections in chickens and their effects on intestinal morphology, immune responses, and cytokine mRNA expression were evaluated in this study.

## 2. Materials and Methods

### 2.1. Preparation of Bacteriocins

PP, a bacteriocin which was originally identified from a *Pediococcus pentosaceus* strain in our previous study, was heterologously expressed and tested for in vitro bacterial inhibition, and was found to be the most effective against *Salmonella*. *S. cerevisiae* strain YTK12 was preserved in yeast extract–peptone–dextrose (YPD; Solarbio, Beijing, China) medium containing 10% glycerol (Solarbio, Beijing, China) in an ultralow-temperature refrigerator at −80 °C. Prior to the experiment, YTK12 was inoculated in YPD liquid medium and incubated aseptically at 30 °C for 20 h. The cultures were then incubated at −80 °C in an ultralow-temperature refrigerator. Subsequently, the culture was transferred to a 5 L fermenter, and fermentation was continued for 2 days. The fermentation broth was centrifuged at 2000× *g* for 10 min at 4 °C, and the supernatant was collected. The supernatant was filtered through a 0.22 μm sterile filter to remove organisms, and the filtrate was stored in a 4 °C refrigerator for backup. The filter-sterilized supernatant, referred to as the crude pediocin PP preparation, was used directly for all subsequent assays and animal administration.

The antimicrobial activity of the crude pediocin PP preparation was determined against *Salmonella* spp. using the agar well diffusion method. Briefly, an overnight culture of the indicator strain was diluted and spread-plated onto Mueller–Hinton agar plates to achieve a final concentration of approximately 3 × 10^7^ CFU per plate. Sterile Oxford cups (Beijing Pro PBS Biotechnology Co., Ltd., (Beijing, China)) (6 mm diameter) were placed on the seeded agar. Subsequently, 200 μL of the crude super-natant preparation was pipetted into each cup. The plates were kept at 4 °C for 4 h to allow pre-diffusion, then incubated at 37 °C for 12 h. The diameter of the resulting in-hibition zones was measured. The preparation used in this study produced a mean inhi-bition zone of 25.6 mm against the target *Salmonella* strain under these conditions. For quantification, the activity titer was defined in Arbitrary Units per milliliter (AU/mL), where 1 AU is defined as the amount of bacteriocin that generates a 1 mm inhibition zone diameter in this assay system. The preparation used in the animal trial had a titer of 1.28 × 10^2^ AU/mL.

### 2.2. Culture and Handling of Salmonella pullorum

*S. pullorum* (CVCC 1800) was obtained from the Microbiological Conservation Center, Beijing Academy of Agriculture and Forestry Sciences. The experimental procedure was as follows: *Salmonella* C1800 was first cultured under aerobic conditions by inoculation into Luria–Bertani (LB) agar medium and incubated at 37 °C while shaking at 200 rpm for 12 h. The culture was transferred to LB liquid medium and incubated for an additional 18 h under the stated conditions. From days 10 to 16 of the experiment, chickens in the challenge group were orally administered a suspension of actively growing *Salmonella* C1800 (1 × 10^7^ colony-forming units [CFU]/mL, 1 mL per chick), whereas control chicks received an equal volume of sterilized LB broth (Qingdao Haibo Biology, Qingdao, China).

### 2.3. Animal Experiments

A total of 45 seven-day-old specific-pathogen-free (SPF) chickens with similar initial body weights were randomly assigned to three treatment groups (*n* = 15): an untreated control group, a *S. pullorum*-challenged group, and a group that received both PP and *S. pullorum* challenge. As illustrated in Figure 1, this design yielded the following three groups: control (CON; negative control, basal diet without *Salmonella* challenge), *Salmonella*-infected CON (SP; positive control, *Salmonella* challenge only), and *Salmonella*-infected + pediocin-treated (SPA) groups. Starting on day 3 (when chickens were 10 days old), chickens in the SP and SPA groups were orally inoculated with 1 mL of actively growing *S. pullorum* culture (1.0 × 10^7^ CFU/mL) for six consecutive days. During the same period, chickens in the CON groups received an equal volume of LB liquid medium. On the 9th day (when chickens were 16 days old), the chickens in the SPA group were orally administered 1 mL of the PP preparation (1.28 × 10^2^ AU) every day for 6 consecutive days until the end of the experiment. During the same period, chickens in the CON groups received an equal volume of YPD liquid medium. Each treatment included 5 replicate cages with 3 chickens per cage. All birds were maintained under uniform environmental conditions in accordance with the Management Guide for chickens. A corn–soybean meal basal diet without antibiotics was provided, which was formulated to meet or exceed the nutrient requirements recommended by the NRC (2004). Chickens were housed under controlled conditions with a 12 h light/12 h dark cycle. Throughout the 15-day experimental period, chickens were group-housed in single-tier stainless-steel cages (1.2 m × 0.9 m × 0.7 m) and provided access to feed and water ad libitum.

### 2.4. Sample Collection

On days 6 and 12 post-infection (corresponding to days 16 and 22 of the experiment), blood samples were collected from 5 chickens per group (1 sample per replicate cage) into coagulant-containing vacuum tubes and centrifuged at 3000 rpm for 10 min. The resulting serum was transferred into 3 aliquots and stored at −20 °C until subsequent analyses. Serum samples were used to determine aspartate aminotransferase (AST), alanine aminotransferase (ALT), Serum creatinine (CREA), and Serum urea (UREA) levels, and other biochemical parameters. Serum C-reactive protein (CRP), serum amyloid A (SAA), tumor necrosis factor (TNF)-α, interleukin (IL)-6, immunoglobulin (Ig)M, and IgG were determined following the manufacturers’ instructions in the corresponding assay kits (Nanjing Jianjian Bioengineering Co., Ltd., Nanjing, China).

On days 6 and 12 post-infection, at each sampling time point, one bird was randomly selected and euthanized from each of the 5 replicate cages per group. This resulted in *n* = 5 chickens per group per time point. Euthanasia was performed via intravenous injection of sodium pentobarbital. Cecal contents from chickens were collected before the end of the trial. Two samples from each replicate were randomly selected, mixed thoroughly, centrifuged, and immediately frozen in dry ice before storage at −80 °C for subsequent analysis of cecal microbiota diversity. Mid-jejunal tissue samples (approximately 2 cm in length) from each bird were excised, snap-frozen in liquid nitrogen, and stored at −80 °C for subsequent mRNA analysis.

### 2.5. Total RNA Extraction and Quantitative Real-Time Polymerase Chain Reaction (PCR) to Determine the Transcript Levels of Relevant Genes in the Jejunum

Total RNA from tissue samples (50 mg) was isolated from snap-frozen jejunal tissues using TRIzol reagent (Invitrogen Life Technologies, Carlsbad, CA, USA). The concentration and purity of total RNA were determined using a microplate reader (Multiskan Sky, 1.00.55, Thermo Fisher Scientific, Waltham, MA, USA) using a 260:280 nm absorbance ratio. Absorbance ratios (optical density [OD]260/OD280) between 1.8 and 2.0 for total RNA samples are considered to be of acceptable purity. First, complementary DNA (cDNA) was synthesized from 1 μg of total RNA using a Fast King reverse-transcription kit (Tiangen, Beijing, China, kit No. KR116) according to the manufacturer’s instructions and stored at −20 °C until further processing. Quantitative real-time PCR was performed using an Applied Biosystems Bio-Rad Real-Time PCR system (Bio-Rad, Carlsbad, CA, USA) and Premix Ex Taq with SYBR Green (Tiangen, China, kit No. FP205). The total reaction mixture was 20 μL, which consisted of 10.0 μL of 2× SYBR PreMix plus, 1.0 μL of cDNA, 0.6 μL of each primer (10 mmol/L), and 7.8 μL RNase-free water. For PCR, the thermal cycling protocol was 15 min at 95 °C, followed by 40 cycles of 10 s each; denaturation at 95 °C, annealing/extension at 60 °C for 34 s; and final melting. Melting curve analysis was performed to monitor the purity of the PCR products. Oligonucleotide primers included chicken *IL-1β*, *IL-6*, *ZO-1*, *claudin*, *occludin*, *MUC2*, *TFF3*, *TGF-β1*, and *TNF-α* (Table 1). Gene expression sequences were designed using Primer Express 5.0 based on the sequences available in public databases. The average gene expression relative to GAPDH was calculated for each sample using the 2^−ΔΔCt^ method.

### 2.6. Histological Assessment

Morphological characteristics of the intestine were assessed by measuring jejunal villi height (VH), crypt depth (CD), and the VH:CD ratio. Briefly, jejunal tissue samples were fixed in 4% paraformaldehyde, and jejunal tissues were subsequently embedded in paraffin and sectioned into 5-μm-thick sections. The prepared sections were stained using hematoxylin and eosin (H&E) and analyzed for jejunal morphometry. VHs and the corresponding CDs of the small intestine were measured using ImageJ software (version 1.54i; National Institutes of Health, USA), and the VH:CD ratio was calculated to complete intestinal histomorphometric analysis.

### 2.7. Microbiota Analysis

Total DNA from the cecal contents of chickens was extracted using a QiagenDNA extraction kit (Qiagen, Shanghai, China), and the extracted DNA was purified and tested for purity and concentration following the manufacturer’s instructions in the kit. The extracted DNA was used as a template for the PCR amplification of the 16S rRNA V3–V4 region of bacteria. PCR products were analyzed using 2% agarose gel electrophoresis, cut gel recovery, and Tris-HCl elution, followed by high-throughput sequencing on the Illumina HiSeq platform, clustering of operational taxonomic units (OTUs), and bioinformatics statistical analysis. Results from OTU clustering analysis were analyzed to determine diversity in colony composition.

### 2.8. Statistical Analysis

Experimental data were initially organized in Microsoft Excel, and one-way analysis of variance was used for analysis in SPSS 23.0. The results are presented as mean ± standard error of the mean (SEM), and differences between groups were compared using *t*-tests and visualized with GraphPad Prism 9 software. Differences with *p* < 0.05, 0.01, and 0.001 were considered statistically significant (*), highly significant (**), and extremely significant (***), respectively.

## 3. Results

### 3.1. Serum Indices

Significant differences in serum biochemical indices were observed among the groups on days 6 and 12 of the intervention (Figure 2). Specifically, on 6 DPI, serum ALT and AST levels were significantly higher (*p* < 0.05) in both the SP and SPA groups compared with those in the CON group. Moreover, CREA and UREA levels were significantly higher than those in the CON group. On 12 DPI, the SP group continued to show significantly elevated ALT and AST (*p* < 0.05) compared with those in the normal control group. Notably, the liver and kidney function indices of PP-treated chickens in the SPA group showed significant improvement, approaching normal levels by the end of the experiment.

As shown in Figure 3, serum IgG and IgM levels were significantly elevated in the SP and SPA groups compared with those in the CON group on 6 DPI of the experiment (*p* < 0.05). As shown in Figure 4, serum IL-6, TNF-α, SAA, and CRP levels were significantly elevated in the SP and SPA groups compared with those in the CON group on 6 DPI of the experiment (*p* < 0.05). After 6 days of PP infusion (on 12 DPI), serum IgG, IgM, TNF-α, IL-6, SAA, and CRP levels in the SPA group were significantly lower than those in the SP group (*p* < 0.05) and returned to near-normal levels by the end of the experiment.

### 3.2. Histological Assessment

Figure 5 and Table 2 demonstrate the effects of PP supplementation on the jejunum morphology of Salmonella-infected SPF chickens. As shown in Figure 5A–C, at 6 DPI, the jejunum villus height (VH) and villus-to-crypt ratio (VH:CD ratio) values were significantly higher in the CON groups than in the SP and SPA groups. At 12 DPI, the villus height and villus-to-crypt ratio in the SP group remained significantly lower than in the CON group. However, after PP intervention, the jejunum VH and VH:CD ratio values in the SPA group were significantly higher than those in the SP group (Figure 5D–F), but did not differ significantly from those in the CON groups.

### 3.3. Real-Time qPCR

The mRNA expression of *claudin-1*, *occludin*, and *Zonula occludens-1* (*ZO-1*) is indicative of the functional status of the intestinal barrier. As shown in Figure 6A–C, on 6 DPI, *Salmonella* infection significantly decreased the expression of the mucosal tight junction (TJ) genes *claudin-1*, *occludin*, and *ZO-1* in the jejunum of the SP and SPA groups. At 12 days post intervention, the expression levels of the TJ genes in the jejunal mucosa increased significantly in the SPA group and approached the values for the control group compared with those noted for the SP group. As shown in Figure 7A,B, at 6 DPI, *Salmonella* infection significantly decreased the expression of the chemical barrier-related gene *Mucin 2* (*MUC2*) in the jejunum of both SP and SPA groups, while markedly increasing *trefoil factor family 3* (*TFF3*) expression. At 12 DPI, the expression of chemical barrier-related genes *MUC2* in the jejunum in the SPA group significantly increased, while the expression of TFF3 significantly decreased and approached the expression noted in the control group when compared with that in the SP group.

As shown in Figure 7C and Figure 8A–D, *Salmonella* infection significantly reduced (*p* < 0.05) the expression of the immune-related gene transforming growth *factor-beta 1* (*TGF-β1*) in the jejunum, while significantly upregulating that of *Toll-like receptor 2* (*TLR2*), *TNF-α*, *IL-1β*, and *IL-6*. After 6 days of intervention, the expressions of jejunal immune-related genes *TLR2*, *TNF-α*, *IL-1β* and *IL-6* in the treatment group decreased significantly, while *TGF-β1* increased significantly and approached those of the control group when compared with that in the SP group.

### 3.4. Microbiota Analysis

To determine the effects of PP supplementation on the intestinal microbiota of *S. pullorum*-infected chickens, high-throughput sequencing of the V3–V4 region of the 16S rRNA gene was conducted using cecal samples (*n* = 5 per group) from the CON, SP, and SPA groups. As shown in Figure 9A,D, significant differences in the α-diversity of the gut microbiota (measured using the Shannon index) were noted among groups on 6 DPI. Specifically, the Shannon indices in the CON groups were significantly higher than those in the SP and SPA groups. However, on 12 DPI, the Shannon index of the SPA group did not differ significantly from that of the CON groups, indicating that PP supplementation partially restored the decreased microbial diversity resulting from *Salmonella* infection.

Phylogenetic analyses revealed distinct compositional profiles of the cecal microbiota at both the phylum and genus levels across treatment groups (Figure 9B,C,E,F). At the phylum level, all groups showed an abundance of Firmicutes and Bacteroidota on 6 DPI. At 6 DPI, the relative abundances of Proteobacteria and Verrucomicrobia were significantly reduced in both the SP and SPA groups compared with those in the control group (*p* < 0.05; see detailed statistical analysis in Table 3). At 12 DPI, Firmicutes remained dominant in all groups, whereas PP supplementation significantly increased the relative abundances of Firmicutes and Actinobacteriota in the SPA group compared with those in the SP group (*p* < 0.05) (Table 3).
Figure 8Effects of *Salmonella pullorum* and pediocin PP on the expression of inflammatory genes in the jejunum of chickens. (**A**,**B**) jejunal pro-inflammatory genes; (**C**,**D**) jejunal anti-inflammatory genes. Vertical bars represent standard errors. *n* = 5 chickens for each treatment. IL-6: Interleukin-6; TGF-β1: transforming growth factor-β1; TNF-α: Tumor Necrosis Factor-alpha; IL-1β: Interleukin-1β treatment. *** denotes *p* < 0.001; ns denotes *p* > 0.05.
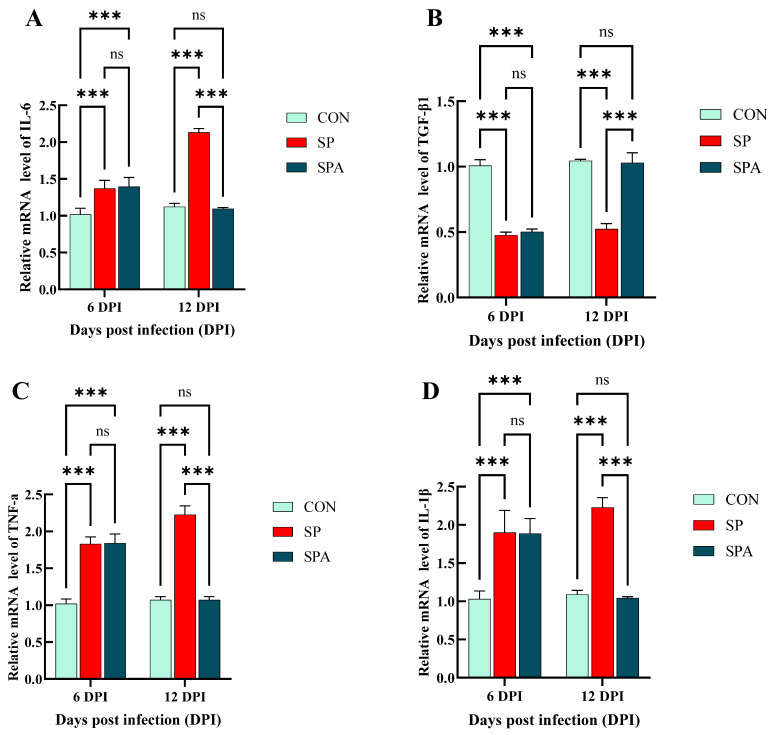


At the genus level, *Subdoligranulum* (32.36%), *Alistipes* (13.47%), and *Faecalibacterium* (10.02%) were predominant across all groups on 6 DPI. On 6 DPI, both SP and SPA groups showed significantly higher abundances of *Erysipelatoclostridium* but lower abundances of *Subdoligranulum* and *Alistipes* compared with those in the CON group. On 12 DPI, the SPA group exhibited higher levels of *Faecalibacterium* and *Butyricicoccus* compared with those in the SP group.
Figure 9Effects of *S. pullorum* and pediocin PP on the diversity of the cecal microbiota in chickens. (**A**,**D**) Shannon indices on 6 DPI and 12 DPI; (**B**,**C**) Relative abundance of the top 20 phyla (**B**) and genera (**C**) in each group on 6 DPI. (**E**,**F**) Relative abundance of the top 20 phyla (**E**) and genera (**F**) in each group on 12 DPI. *n* = 5 chickens for each treatment.
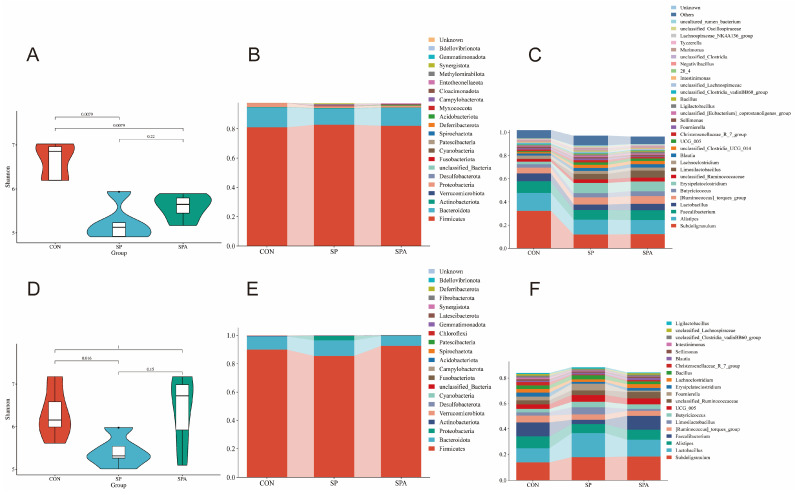


## 4. Discussion

Chicken dysentery is an acute or chronic infectious disease caused by *S. pullorum*. Pullorum disease, caused by *Salmonella pullorum*, primarily affects chicks aged 2–3 weeks. Infected chicks often exhibit huddling together for warmth, lethargy, loss of appetite, and dehydration, with characteristic white, paste-like droppings around the vent [21]. Cheng [22] found that within 5 to 7 days post-infection with *Salmonella* Pullorum, all chickens exhibited symptoms of labored breathing and mental depression, with some mortality occurring. The study also revealed a significant upregulation in the expression of inflammatory factors—tumor necrosis factor-alpha (TNF-α), interleukin-1β (IL-1β), IL-6, and IL-8 in both lung and spleen tissues. Furthermore, histopathological examination indicated swelling of the lung tissue accompanied by distinct lesions. Previous studies have shown that S. pullorum infection significantly reduces the average body weight of Hongguang-Black 1-day-old breeder chicks [23]. This disease mainly affects poultry and is particularly harmful to young chickens [24]. Urea is an end product of protein metabolism that is primarily synthesized in the liver and excreted by the kidneys [25]. CREA, a product of muscle metabolism, is also mainly excreted via the kidneys. Moreover, CREA is commonly used as an auxiliary indicator when evaluating renal function in poultry in clinical settings [26]. A previous study has reported that *Salmonella* infection significantly increases serum alanine aminotransferase, aspartate aminotransferase, and alkaline phosphatase activity, as well as UREA and CREA level [27]. Our findings revealed that *S. pullorum* infection raised the levels of UREA and CREA, while its level was close to the control group when supplemented with pediocin PP, suggesting that PP supplementation could alleviate dysentery in chickens after *S. pullorum* infection. ALT and AST are key enzymes that catalyze transamination reactions between amino acids and keto acids. In mammals, these enzymes are released into the circulation when hepatocyte membranes are disrupted due to cellular injury or metabolic dysfunction. The extent of increase in serum ALT levels is positively correlated with the severity of hepatocellular damage, making it a reliable biomarker in assessing liver function [28,29]. Findings suggest that elevated ALT levels following *Salmonella* infection may reflect the capacity of the pathogen in colonizing hepatic tissues and inducing hepatocellular damage [30]. Previous studies have reported that *S. enteritidis* (SE) infections could significantly increase ALT activity in 1- and 7-day-old chicks within the SE-infected subgroup (S-II) compared with those in the Intebiot-fed, SE + sage extract-fed, and negative control groups [31]. Similarly, among *S. pullorum*-infected chicks, the SP-challenged subgroup exhibited hepatic hemorrhage and focal necrosis compared with those in the CON and treatment subgroups that received 0.5%, 1%, or 1.5% acetic acid. Notably, supplementation with 1% acetic acid markedly reduced the severity and frequency of these pathological lesions [32]. Our findings revealed a significant increase in ALT and AST levels in both infected groups (SP and SPA) on 6 DPI. Similar findings have been reported in 1-day-old Salmonella-challenged SPF Lohmann broiler chicks, where AST activity was significantly increased at 1, 3, and 5 days post-infection [33]. Therefore, our present results suggest that Salmonella infection exerts negative physiological effects on poultry, whereas supplementation with PP may confer hepatoprotective benefits [34,35,36].

*S. pullorum* infection induces inflammatory responses by releasing cytokines and disrupting the intestinal barrier, thereby altering systemic immune parameters [37]. In our study, *S. pullorum* challenge markedly increased serum IgG and IgM levels, consistent with previous findings showing Salmonella-induced humoral immune activation [38]. Colonization of the gastrointestinal tract by Salmonella can compromise the intestinal barrier and activate mucosal immune pathways, triggering TNF-α and IL-6 release and promoting systemic inflammation [39]. In the current study, *S. pullorum* challenge markedly increased IL-6 levels, whereas PP supplementation significantly reduced TNF-α and IL-6 levels, indicating the anti-inflammatory effect of PP. SAA and CRP are acute-phase proteins and sensitive markers of inflammation [40]. In this study, *S. pullorum* infection significantly elevated serum SAA and CRP levels, likely reflecting the systemic inflammatory response triggered by *Salmonella*. Pathogen invasion activates the innate immune system in the host, stimulating hepatic synthesis of acute-phase proteins such as SAA and CRP, whose elevated concentrations indicate the severity of infection-induced inflammation [41,42]. PP supplementation reduced SAA and CRP levels, suggesting that PP could alleviate SP-induced intestinal barrier disruption and systemic inflammation.

Our findings also demonstrated that *Salmonella* infection significantly decreased the jejunal expression of the immunomodulatory cytokine *TGF-β1* while markedly increasing the expression of *TLR2*, *TNF-α*, *IL-1β*, and *IL-6*. These findings highlight the complex regulatory effects of *Salmonella* on the intestinal immune response and provide insights into its pathogenic mechanisms. *TGF*-*β1* is a multifunctional cytokine crucial for maintaining immune homeostasis, suppressing inflammation, and promoting tissue repair [43]. The downregulation of *TGF-β1* noted in our study may weaken immune regulation in the host, resulting in excessive inflammatory activation, consistent with previous studies that have also suggested that *Salmonella* promotes inflammation by inhibiting the expression of *TGF-β1* [44]. This imbalance may further aggravate intestinal inflammation, facilitating *Salmonella* colonization and tissue invasion. Conversely, the expressions of *TLR2*, *TNF-α*, *IL-1β*, and *IL-6* were significantly upregulated after infection. *TLR2*, a member of the pattern recognition receptor family, detects pathogen-associated molecular patterns and initiates innate immune signaling [45]. Elevated *TLR2* expression indicates activation of the innate immune system of the host, whereas the upregulation of *TNF-α*, *IL-1β*, and *IL-6* reflects an intense inflammatory response that may result in tissue injury and compromise barrier integrity [46]. Remarkably, on 12 DPI, the expression of these immune-related genes was almost similar to that in the control group. This finding suggests that PP intervention can restore intestinal immune balance by upregulating *TGF-β1* levels and downregulating those of *TLR2*, *TNF-α*, *IL-1β*, and *IL-6*. Enhanced *TGF-β1* expression may suppress excessive inflammation, whereas normalization of the levels of pro-inflammatory cytokines promotes tissue recovery. These findings were consistent with previous reports that probiotics and bioactive compounds ameliorate intestinal inflammation by modulating immune-related gene expression [47,48]. Overall, *Salmonella* infection disrupts intestinal immune homeostasis via differential regulation of pro- and anti-inflammatory mediators, whereas PP supplementation re-establishes the immune balance and promotes mucosal healing.

Maintaining the integrity of the intestinal morphology is essential for effective nutrient absorption owing to its crucial role in preserving the intestinal barrier. Intestinal villi are vital structural components of the small intestine that are responsible for nutrient uptake [49]. Previous studies have reported that *S. pullorum* infections severely damage intestinal villi and decrease nutrient absorption [50]. In our study, *S. pullorum* infection markedly impaired the jejunal VH and VH:CD ratio, which may have contributed to the observed decline in growth performance. Similarly, previous studies have demonstrated that *Lactobacillus paracasei* KL1 and *L. plantarum* can mitigate histopathological lesions in the intestine and preserve villus integrity in *S. dysenteriae*-infected chicks [51]. Supplementation with bacteriocins has been shown to improve intestinal histomorphology in Japanese quail [52]. Consistent with these findings, supplementing the diet with PP in our study enhanced VH, alleviated *S. pullorum* infection–induced jejunal damage, and promoted faster epithelial regeneration and improved nutrient absorption. Notably, after PP intervention, the SPA group exhibited a significantly increased jejunal VH and VH:CD ratio, with values comparable to those in the CON groups. These results indicate that PP could effectively alleviate *S. pullorum*-induced intestinal morphological damage to near-normal levels. This finding aligns with that reported previously that probiotics and their metabolites strengthen the intestinal barrier by promoting epithelial proliferation and differentiation [53], thereby enhancing resistance to pathogenic infection. In conclusion, PP supplementation could effectively mitigate *S. pullorum*-induced intestinal injury and promote intestinal recovery.

TJs of the intestinal epithelium are crucial for maintaining barrier integrity, with *occludin*, *claudin-1*, and *ZO-1* serving as core structural proteins [54]. The expression of these proteins is positively correlated with intestinal barrier function, and their downregulation can compromise TJ integrity, increasing intestinal permeability [55,56]. *Salmonella* infection significantly decreases the expression of barrier function-related genes and increases epithelial permeability [57,58]. Bu et al. [59] have reported that bacteriocin-producing *L. plantarum* can markedly upregulate intestinal *ZO-1* and *claudin-1* expression in mice, thereby exerting an enteroprotective effect. Similarly, Enterococcus species can attenuate diet-induced intestinal inflammation and preserve epithelial integrity [60,61]. In the current study, *S. pullorum* infection significantly downregulated the mRNA expression of *occludin* and *claudin-1* in the jejunum of broilers. However, PP supplementation reversed this effect by upregulating *claudin-1* expression, consistent with previous observations. These findings collectively suggest that PP mitigates *S. pullorum* infection-induced impairment of intestinal barrier function by modulating the expression of TJ proteins.

*MUC2* and *TFF3* are key regulators of the intestinal chemical barrier and play a crucial role in maintaining intestinal homeostasis and defending against pathogenic infection [62,63]. In this study, *S. pullorum* infection significantly downregulated *MUC2* expression in the jejunum while markedly upregulating that of *TFF3*, revealing the dual regulatory effect of *Salmonella* on the intestinal chemical barrier. *MUC2* is the primary structural component of the intestinal mucus layer. It is secreted by goblet cells to form a protective barrier that prevents pathogens from directly coming into contact with epithelial cells [64]. The downregulation of *MUC2* observed in our study likely reduced the thickness of the mucus layer and weakened the intestinal defense against *Salmonella,* consistent with previous studies that have reported that pathogenic infections disrupt barrier integrity by inhibiting *MUC2* expression [65]. Reduced *MUC2* expression may also exacerbate intestinal inflammation, thereby facilitating *Salmonella* colonization and invasion. In contrast, *TFF3* expression is significantly upregulated after *Salmonella* infections. *TFF3*, a member of the *trefoil factor* family, plays an important role in mucosal repair and regulating inflammation [66]. Its upregulation may represent a compensatory host response aimed at promoting epithelial cell migration, tissue repair, and attenuation of inflammation. However, increased *TFF3* expression has been associated with chronic inflammation and pathological conditions [67]. Notably, PP supplementation normalized the expression of both *MUC2* and *TFF3*, suggesting its ability to restore the chemical barrier function of the intestine by modulating these genes. Upregulation of *MUC2* is associated with mucus layer reconstruction and reinforcement of the intestinal physical barrier, whereas restoring *TFF3* expression promotes epithelial repair and mitigates inflammation. These findings align with previous findings that probiotics and certain pharmacological agents can enhance the integrity of the intestinal barrier by regulating the expression of *MUC2* and *TFF3* [68,69]. Overall, PP supplementation effectively alleviated *S. pullorum*-induced disruption of the intestinal chemical barrier by restoring the expression of *MUC2* and *TFF3*.

The cecal microbiota plays a crucial role in nutrient absorption, host immunity, and intestinal defense [70,71]. However, infections by intestinal pathogens disrupt microbial homeostasis, leading to intestinal inflammation and impaired growth performance [72]. In our study, the effects of PP supplementation on the intestinal microbial composition of chickens infected with *S. pullorum* were systematically analyzed using 16S rRNA gene sequencing. *Salmonella* infection significantly reduced the α-diversity of the intestinal microorganisms, which was partially restored after PP intervention. At the phylum level, Firmicutes and Bacteroidetes were predominant in all experimental groups, consistent with previous reports on avian intestinal microbiota composition [73]. PP supplementation significantly increased the relative abundance of Actinobacteria, suggestive of its probiotic effect by modulating specific microbial taxa. At the genus level, PP treatment markedly increased the relative abundance of *Faecalibacterium* and *Lactobacillus*, both of which are known to exert anti-inflammatory and probiotic effects [74,75]. Furthermore, the relative abundance of *Faecalibacterium* and *Butyricicoccus*, genera strongly associated with short-chain fatty acid production and intestinal health, was significantly elevated in the PP-treated groups [76]. Overall, PP supplementation partially restored the microbial diversity that was disrupted by *Salmonella* infection and promoted the proliferation of beneficial bacteria, thereby re-establishing the intestinal microbial balance and improving gut health.

## 5. Conclusions

Our study provides preliminary evidence that supplementation with PP alleviates the inflammatory response in *S. pullorum*-challenged chickens. Moreover, PP supplementation mitigated the subclinical *Salmonella* infection by improving intestinal morphology and barrier function and optimizing the composition of the cecal microbiota. Our findings offer a theoretical basis for the protective role of PP in reducing *S. pullorum* infections and highlight its potential as a therapeutic strategy in controlling avian salmonellosis. Overall, this study not only elucidates the underlying mechanisms by which PP mitigates *S. pullorum* infections but also offers a valuable tool for therapeutic intervention.

## Figures and Tables

**Figure 1 microorganisms-14-00018-f001:**
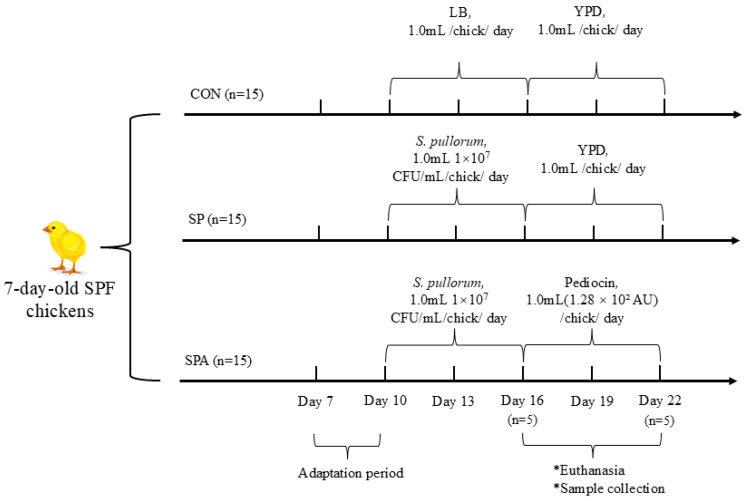
Schematic overview of the experimental design. Forty-five 7-day-old specific-pathogen-free (SPF) chickens were randomly assigned to three groups: control (CON), *Salmonella pullorum* infection (SP), and infection plus pediocin PP treatment (SPA). Starting on Day 3 (10 days of age), chickens in the SP and SPA groups were orally inoculated with *S. pullorum* (1.0 × 10^7^ CFU/mL/day) for 6 consecutive days, while CON group received sterile LB medium. From Day 9 (16 days of age), chickens in the SPA group received daily oral administration of pediocin PP preparation (1.28 × 10^2^ AU/day) for 6 days, while the CON and SP groups received sterile YPD medium. Key sampling time points are indicated.

**Figure 2 microorganisms-14-00018-f002:**
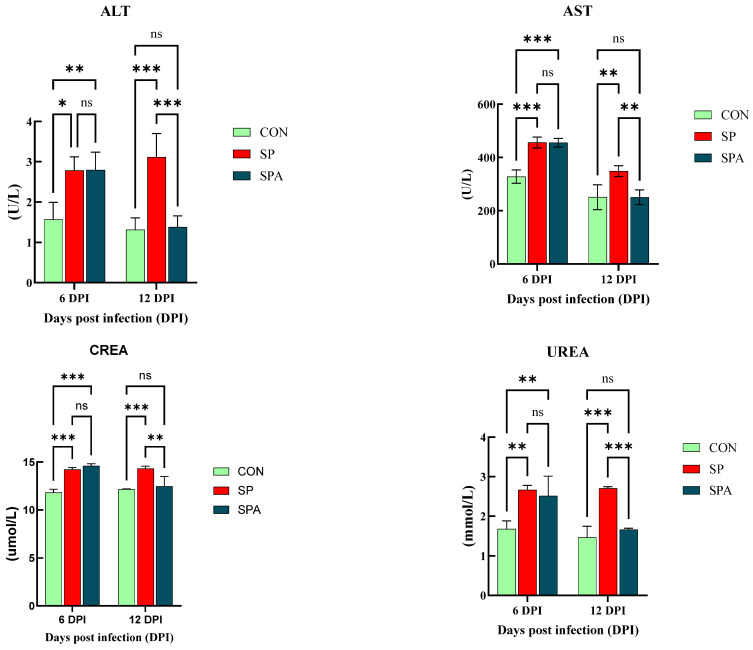
Effect of *Salmonella pullorum* and pediocin PP on serum liver and renal function in chickens. ALT: Alanine Aminotransferase; AST: Aspartate Aminotransferase; CREA: Serum creatinine; UREA: Serum urea; UREA: Urea uric acid. * denotes *p* < 0.05; ** denotes *p* < 0.01; *** denotes *p* < 0.001; ns denotes *p* > 0.05.

**Figure 3 microorganisms-14-00018-f003:**
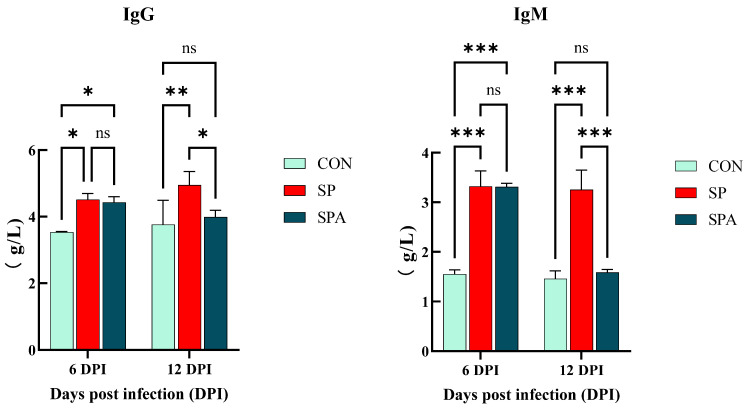
Effect of *Salmonella pullorum* and pediocin PP on serum immunity of intestinal damage in chickens. IgG: ImmunogIobuIin G; IgM: ImmunogIobuIin M. * denotes *p* < 0.05; ** denotes *p* < 0.01; *** denotes *p* < 0.001; ns denotes *p* > 0.05.

**Figure 4 microorganisms-14-00018-f004:**
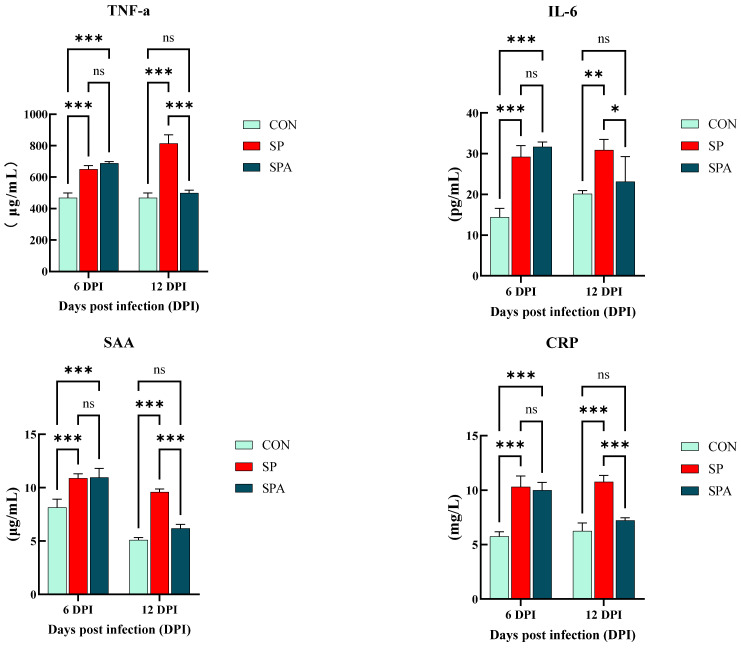
Effect of Salmonella pullorum and pediocin PP on serum pro-inflammatory factors and markers of intestinal damage in chickens. TNF-α: tumor necrosis factor-α; IL-6: Interleukin-6; SAA: serum amyloid A; CRP: C-reactive protein.* denotes *p* < 0.05; ** denotes *p* < 0.01; *** denotes *p* < 0.001; ns denotes *p* > 0.05.

**Figure 5 microorganisms-14-00018-f005:**
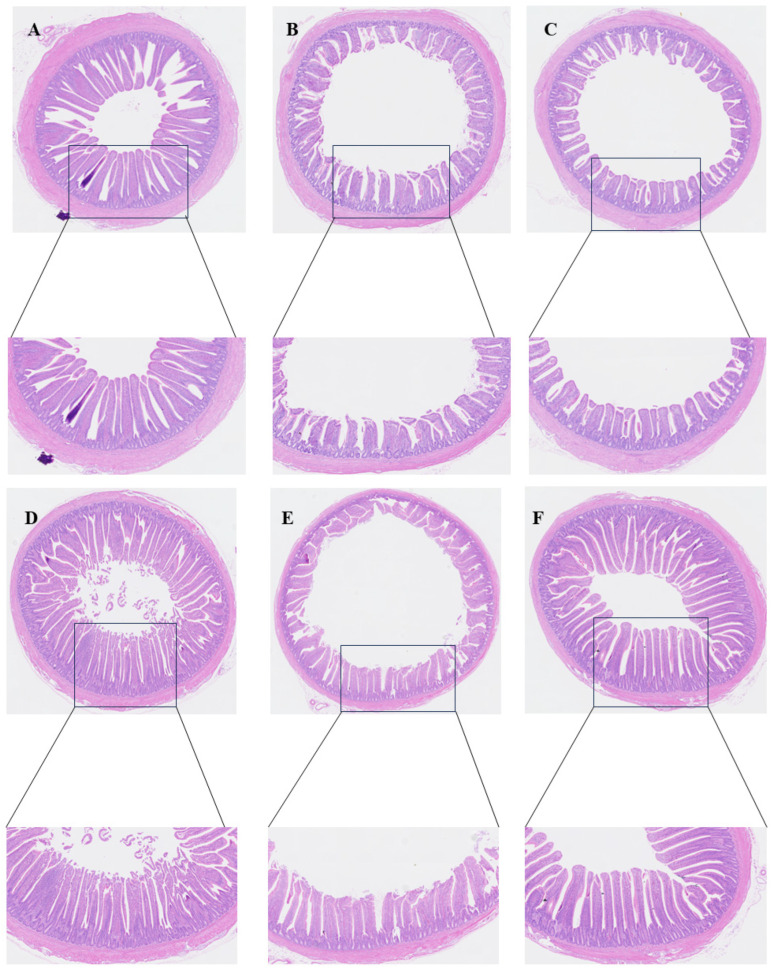
Effect of pediocin PP intervention on the jejunum morphology of chickens with *S. pullorum*. Magnification was 5× and 10×, *n* = 5 chickens per treatment. Representative photomicrographs of jejunal sections from chickens. (**A**–**C**) Sections from the CON, SP, and SPA groups at 6 DPI, respectively. (**D**–**F**) Sections from the CON, SP, and SPA groups at 12 DPI, respectively.

**Figure 6 microorganisms-14-00018-f006:**
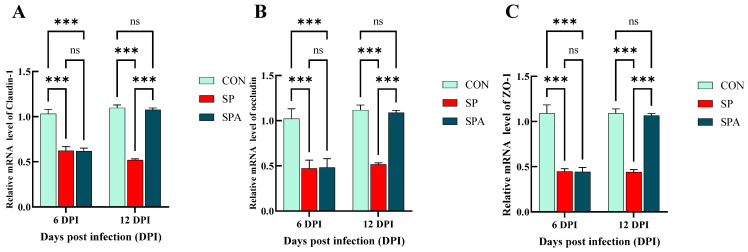
Effects of *Salmonella pullorum* and pediocin PP on the expression of jejunal barrier–related genes in chickens. (**A**–**C**) Physical barrier genes in the jejunum. Vertical bars represent standard errors. *n* = 5 chickens for each treatment. ZO-1: zonula occludens-1. *** denotes *p* < 0.001; ns denotes *p* > 0.05.

**Figure 7 microorganisms-14-00018-f007:**
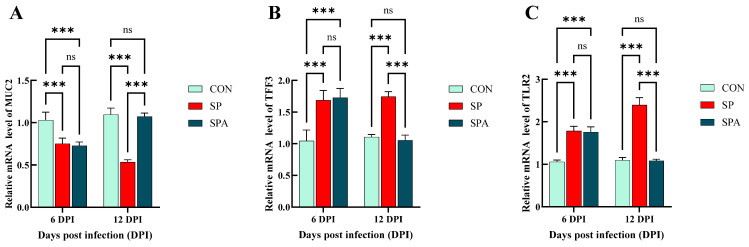
Effects of *Salmonella pullorum* and pediocin PP on the expression of jejunal barrier–related genes in chickens. (**A**,**B**) Chemical barrier genes in the jejunum. (**C**) Immune barrier genes in the jejunum. Vertical bars represent standard errors. *n* = 5 chickens for each treatment. MUC2: Mucin 2; TFF3: Trefoil factor family 3; TLR2: Toll Like Receptor 2. *** denotes *p* < 0.001; ns denotes *p* > 0.05.

**Table 1 microorganisms-14-00018-t001:** Primer sequences.

Target Gene	Forward/Reverse	Gene Bank Accession Number
*GAPDH*	F: CAACCCCCAATGTCTCTGTT	NC_052532.1
	R: TCAGCAGCAGCCTTCACTAC	
*Claudin-1*	F:GATGCGGATGGCTGTCTTTGGT	NP_001013629
	R:GGCTGGGTGGGTAGGATGTTTC	
*Occludin*	F:TGTGGGTTCCTCATCGTCATCC	NM_205128.1
	R:TCTCCGAGTAGGCAAGCGTGG	
*ZO-1*	F:TCTCCCTAAAGGCGAAGAAGTAACC	XM_015278980.1
	R:ATAGCGTGTCCACAACCCGAAAC	
*CLL5*	F: CAACCGTGTGCTGCTTCAACTAT	NC_052550.1
	R:CCTTCCTGGTGATGAACACAACTG	
*MUC2*	F:GAGCCACCTCCTAAACCCACCTG	NM_001318434.1
	R:GCAATCACACTCCCAATGCCAAC	
*TFF3*	F:TGCTTGCTCTTTGTCATCCTTGC	XM_416,743.4
	R:GCTGCTGAGATTCCTGGGGG	
*TLR2*	F: GAGGGGCACAGGTTGGGAG	NW_026294743.1
	R: CAGATGTCTTTCGTGGGGGC	
*TNF-α*	F: AACTATCCTCACCCCTACCCTGTC	NM_204267.2
	R: GGGCGGTCATAGAACAGCACT	
*IL-1β*	F: CCGCTACACCCGCTCACAGT	XM_015297469.2
	R: TGCCGCTCATCACACACGAC	
*IL-6*	F: GCCTGTTCGCCTTTCAGACCTAC	NC_052533.1
	R: TCGGGATTTATCACCATCTGCC	
*TGF-β1*	F: CAAGGATCTGCAGTGGAAGTGG	NC_052569.1
	R: CGTAGTAAATGATGGGGAGGGG	

**Table 2 microorganisms-14-00018-t002:** Effect of PP intervention on the jejunum morphology of chickens with *S. pullorum* of broiler chickens at 6 DPI and 12DPI.

		Treatment Group				
	Site	CON	SP	SPA	SEM	*p*-Value
6 DPI	Villus height (μm)	1209.45 ^a^	1113.11 ^b^	1109.40 ^b^	18.80	0.042
	Crypt depth (μm)	190.77	194.67	195.67	2.28	0.666
	Villus height: crypt depth	6.40 ^a^	5.75 ^b^	5.67 ^b^	0.13	0.049
12 DPI	Villus height (μm)	1222.96 ^a^	1128.78 ^b^	1212.20 ^a^	15.76	0.022
	Crypt depth (μm)	196.50	194.45	196.07	2.16	0.925
	Villus height: crypt depth	6.22 ^a^	5.81 ^b^	6.18 ^a^	0.04	0.001

Means of cross- of the jejunum from each bird, with five birds per treatment group, and 6 measurements of each villus height, crypt depth. ^a,b^ Means with different superscripts within a row differ significantly (*p* < 0.05).

**Table 3 microorganisms-14-00018-t003:** Effect of PP intervention on the jejunal phylum levels of chickens with S. pullorum of broiler chickens at 6 DPI and 12DPI.

	Phylum Level (%)	Treatment Group				
		CON	SP	SPA	SEM	*p*-Value
6DPI	Proteobacteria	0.123 ^a^	0.056 ^b^	0.056 ^b^	0.0114	0.01
	Verrucomicrobia	0.0060 ^a^	0.002 ^b^	0.002 ^b^	0.0001	0.01
12DPI	Firmicutes	91.000 ^a^	84.000 ^b^	91.000 ^a^	1.2401	0.01
	Actinobacteriota	0.254 ^a^	0.165 ^b^	0.235 ^a^	0.0141	0.01

^a,b^ Means with different superscripts within a row differ significantly (*p* < 0.05).

## Data Availability

The sequences of metagenomes presented in this study are openly available in GenBank under BioProject PRJNA1354233.

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
