# Peer review of "Dietary Pediocin Supplementation Restores Intestinal Barrier Function and Microbiota Balance in Salmonella-Infected Specific-Pathogen-Free Chickens"

_microorganisms, 2025, doi:10.3390/microorganisms14010018_

Round 1
Reviewer 1 Report
Comments and Suggestions for Authors
Overall Assessment & Recommendation: This is a well-designed, informative, and clinically relevant study that addresses an important issue in poultry health and food safety. The experimental design is sound, the data are generally robust, and the findings support the conclusion that pediocin (PP) has protective effects against S. pullorum infection. The manuscript is well-written but requires significant editorial polishing, correction of numerous errors, and clarification of several key points. In its current form, it is not yet ready for acceptance.
Detailed Critique by Section
Abstract:
Line 15: "This aim of this study was to investigate the effects of pediocin (PP) on intestinal barrier function, renal injury, and immune regulation were evaluated..." Grammar Error. Use either "The aim of this study was to investigate the effects..." or "In this study, the effects of... were evaluated."
Line 21-22: "the expression of trefoil factor 3, Toll-like receptor 2, TNF-α, IL-1β, and IL-6" – The list mixes genes typically upregulated and downregulated. The abstract states these were "significantly elevated," which contradicts the results for TFF3 (Fig 7B shows upregulation) and the later text about gene expression. This is a major inconsistency. The abstract must accurately reflect the results.
Line 25-26: "...whereas these elevations persisted in the SP group at 12 DPI, they were restored in the SPA group..." The antecedent for "these elevations" is unclear. Rephrase for clarity.
Lines 29-30: "PP increased the relative abundance of Proteobacteria and Verrucomicrobia" – This is contradicted by the results text (Lines 302-303), which states these phyla were reduced at 6 DPI. The intended meaning (that PP restored them) is not clearly stated. Inconsistency.
The abstract effectively summarizes key findings but needs precise language.
- Introduction (Lines 37-85)
Lines 79-81: "In our laboratory, a Saccharomyces cerevisiae strain YTK12 heterologously expressing pediocin (PP) was originally constructed..." This is a crucial methodological point that belongs in the Materials and Methods (Section 2.2), not the Introduction. The Introduction should state the purpose of using this construct.
Line 82: "Pediococcus pentosaceus-derived bacteriocins" – This is confusing because Section 2.2 says PP was discovered from P. pentosaceus but expressed in yeast. Be consistent: is PP from P. pentosaceus or P. acidilactici (mentioned in Line 70)? Clarify the source.
- Materials and Methods (Lines 86-203)
2.2. Preparation of bacteriocins:
Lines 93-94: "PP, a bacteriocin discovered from Pediococcus pentosaceus strain in the pre-laboratory..." Move the discovery statement here from the Introduction.
Lines 96-104: The description of fermentation and processing is vague. What was the final product? Crude supernatant? Was the pediocin concentrated or purified? What was the titer or activity (e.g., Arbitrary Units/mL)? This is a critical omission. The dose ("1 mL of PP") is meaningless without knowing its bioactive content.
2.3. Culture and handling: Clear.
2.4. Animal experiments:
Line 118: "two of which was challenged" – Grammar. Change to "two of which were challenged" or, better, rephrase the entire sentence for clarity. The group description is confusing.
Figure 1: The schematic is helpful but not referenced in the text (no callout like "Figure 1").
The timeline of infection (days 10-16) and PP treatment (starting day 16) is clear but unusual. PP was administered after the infection protocol ended. This should be explicitly justified in the discussion as a therapeutic rather than preventive model.
2.5. Sample collection: Clear.
2.6. RNA extraction and qPCR:
Table 1: The primer for `CLL5` seems erroneous (Forward and Reverse sequences are identical). Is this a typo? Also, `TNF-α` and `TGF-β1` primers are listed, but their accession numbers are from different databases (MF000729.1 vs NC_052569.1). This is acceptable but should be consistent.
The housekeeping gene is listed as `GAPDH` in the table but referred to as `β-actin` in the text (Line 175). Major inconsistency. Use one consistently.
2.7-2.9: Standard and acceptable.
- Results (Lines 204-312)
3.1. Serum indices:
Figures 2, 3, and 4 are well-presented. However, the y-axis label in Figure 2 for CREA says "Cera creatinine" – Typo. Should be "Serum creatinine".
Line 209: "CERA" is used in the text, but the figure label and legend use "CREA". Inconsistency. Standardize to "CREA" or "Cr".
3.2. Histological assessment:
Line 235: "Figure 5 and table 3" – Error. It should be "Table 2" (found at the end of the manuscript). Table 2 is misplaced; it should be in the results section after Figure 5.
Figure 5: Images are small. Higher magnification insets would help visualize villus/crypt differences.
3.3. Real-time qPCR:
Figure 7A Label Error: The legend says "SPA" but the figure bar is labeled "SPA" for the orange bar. The text in the figure says "CON, SPA, SPA". This is incorrect. It should be CON, SP, SPA.
The results are clearly described, but the narrative would benefit from stating the direction of change (up/down) more explicitly.
3.4. Microbiota analysis:
Lines 302-303: Text says Proteobacteria and Verrucomicrobia were "significantly reduced" at 6 DPI in SP and SPA vs. CON. However, Figure 9B does not visually support a major reduction for these phyla. The statistical analysis must be shown (e.g., in a supplementary table). This is a potential inconsistency between the text and the figure.
Line 305: "PP supplementation significantly increased the relative abundances of Firmicutes and Actinobacteriota" – Check Figure 9E; the increase in Actinobacteriota is visible, but Firmicutes appear similar between SP and SPA. Again, statistics are needed.
The term "Micrococcaceae" (Line 460) is a family, not a phylum. The intended phylum is likely Actinobacteriota.
- Discussion (Lines 313-469)
Generally thorough and relates findings well to existing literature.
Line 320: "UREA levels were lower in the SP and SPA groups..." This contradicts Figure 2, which shows UREA levels are higher in SP and SPA at 6 DPI. Major factual error. Please correct.
Lines 359-360: "Pathogen invasion activates the innate immune system of the host" – Grammar. "of the host" -> "in the host".
The discussion effectively weaves together serum markers, histology, gene expression, and microbiota.
Key Omission: There is no dedicated "Limitations" section. Limitations must be explicitly discussed. For example:
- The dose of PP is not quantified in bioactive units.
- PP treatment began after infection, which may not reflect real-world prophylactic use.
- The use of SPF chickens may not fully represent commercial flocks with complex microbiota.
- The mechanisms linking microbiota changes to improved barrier function are correlative, not causative, in this study.
- General Issues & English Corrections
Grammar & Typos: Frequent throughout. Examples:
Line 13: "zhdy203@126.com (DY. Zhang, Tel.: +86 10 81 127 468)." – Phone number formatting is odd.
Line 49: "microbiota of the host" -> "host microbiota".
Line 200: "standard error of the mean (SEM)" – formatting error with extra spaces.
Line 384: "are crucial in maintaining" -> "are crucial for maintaining".
Line 399: "against pathogenic infections" -> "against pathogenic infection".
In figures: "ImmunogIobuIin" (odd capitalization).
Formatting: The document has numerous formatting artifacts from copying (e.g., "", "" symbols, misplaced line numbers in the text body).
Reference Callouts: Some references are called out as `[23]` in the results (Lines 220, 221), which is inappropriate. These should be used only in the discussion to compare results. Results should present your data without references.
The study has merit and valuable findings. However, due to the presence of major inconsistencies, omission of key methodological details, and the need for extensive linguistic correction, the manuscript must undergo major revisions before it can be considered for publication. I recommend rejection in the current form with a detailed review to guide a thorough resubmission.
Author Response
Dear Reviewer,
Thank you for your letter and the reviewers’ constructive comments on our manuscript “Dietary Pediocin supplementation restores intestinal barrier function and microbiota balance in Salmonella-infected specific-pathogen-free chickens” (ID: ISSN 2076-2607). We have carefully considered all suggestions and have revised the manuscript accordingly. Our point-by-point responses are provided below:
Comments 1: Line 15: " Thank you for your valuable comments. Line 15 has been revised as: “In this study, the effects of pediocin (PP) on intestinal barrier function, renal injury, and immune regulation were evaluated in Salmonella pullorum infected chickens.”
Response 1: Thank you for your valuable comments. Line 15, “The purpose of this study is to examine” has been revised and the manuscript has been language edited to correct grammatical errors and revise this sentence.
Comments 2: Line 21-22: "the expression of trefoil factor 3, Toll-like receptor 2, TNF-α, IL-1β, and IL-6" – The list mixes genes typically upregulated and downregulated. The abstract states these were "significantly elevated," which contradicts the results for TFF3 (Fig 7B shows upregulation) and the later text about gene expression. This is a major inconsistency. The abstract must accurately reflect the results.
Response 2: Thank you for your valuable comments. Line 19-22 have been revised as: “The results showed that S. pullorum infection significantly affected the levels (P < 0.05) of renal (CREA, UREA), hepatic (ALT, AST), immunological (IgG, IgM), inflammatory (TNF-a, IL-6, SAA, CRP) parameters, and the expression of trefoil factor 3, Toll-like receptor 2, TNF-a, IL-1b, and IL-6.”
Comments 3: Line 25-26: "...whereas these elevations persisted in the SP group at 12 DPI, they were restored in the SPA group..." The antecedent for "these elevations" is unclear. Rephrase for clarity.
Response 3: Thank you for your valuable comments. Changes have been made to Lines 25-28: “In the SP group, the parameter alterations observed at 6 DPI compared to the CON group persisted until 12 DPI. In contrast, in the SPA group, these parameters returned to levels comparable to those of the CON group after 6 days of PP treatment.
Comments 4: Lines 29-30: "PP increased the relative abundance of Proteobacteria and Verrucomicrobia" – This is contradicted by the results text (Lines 302-303), which states these phyla were reduced at 6 DPI. The intended meaning (that PP restored them) is not clearly stated. Inconsistency.
Response 4: Thank you for the valuable comments from the reviewers. Line 29-33 have been changed as: “At the phylum level, S. pullorum infection significantly reduced the relative abundances of Proteobacteria and Verrucomicrobia. PP treatment increased the abundances of Firmicutes and Actinobacteria, while also restoring the abundances of Proteobacteria and Verrucomicrobia to some extent.”
Comments 5: Lines 79-81: "In our laboratory, a Saccharomyces cerevisiae strain YTK12 heterologously expressing pediocin (PP) was originally constructed..." This is a crucial methodological point that belongs in the Materials and Methods (Section 2.2), not the Introduction. The Introduction should state the purpose of using this construct.
Response 5: Thank you for your valuable comments. In the “Introduction”section, Line 87-89 have been changed as: “Previously, an engineered strain of Saccharomyces cerevisiae producing pediocin was successfully constructed. This strain exhibited excellent antibacterial properties in vitro and had a particularly strong antagonistic effect against Salmonella.”
Comments 6: Line 82: Pediococcus pentosaceus-derived bacteriocins" – This is confusing because Section 2.2 says PP was discovered from P. pentosaceus but expressed in yeast. Be consistent: is PP from P. pentosaceus or P. acidilactici (mentioned in Line 70)? Clarify the source
Response 6: Thank you for your valuable comments. pediocin PP was originally identified from a Pediococcus pentosaceus, it was due to our writing mistake. In the section of “2.2 Preparation of bacteriocins”, it has been changed as: “PP, a bacteriocin which was originally identified from a Pediococcus pentosaceus strain in our previous study”.
Comments 7: Lines 93-94: "PP, a bacteriocin discovered from Pediococcus pentosaceus strain in the pre-laboratory..." Move the discovery statement here from the Introduction.
Response 7: Thank you for your valuable comments. Lines 103-104 have been changed as: “Previously, an engineered strain of Saccharomyces cerevisiae producing pediocin was successfully constructed. This strain exhibited excellent antibacterial properties in vitro and had a particularly strong antagonistic effect against Salmonella.”
Comments 8: Lines 96-104: The description of fermentation and processing is vague. What was the final product? Crude supernatant? Was the pediocin concentrated or purified? What was the titer or activity (e.g., Arbitrary Units/mL)? This is a critical omission. The dose ("1 mL of PP") is meaningless without knowing its bioactive content.
Response 8: Thank you for your valuable comments. Line 116-128 have been changed as: “The antimicrobial activity of the crude pediocin PP preparation was determined against Salmonella spp. using the agar well diffusion method. Briefly, an overnight culture of the indicator strain was diluted and spread-plated onto Mueller–Hinton agar plates to achieve a final concentration of approximately 3×10⁷ CFU per plate. Sterile Oxford cups (6 mm diameter) were placed on the seeded agar. Subsequently, 200 μL of the crude super-natant preparation was pipetted into each cup. The plates were kept at 4°C for 4 hours to allow pre-diffusion, then incubated at 37°C for 12 hours. The diameter of the resulting in-hibition zones was measured. The preparation used in this study produced a mean inhibition zone of 25.6 mm against the target Salmonella strain under these conditions. For quantification, the activity titer was defined in Arbitrary Units per milliliter (AU/mL), where 1 AU is defined as the amount of bacteriocin that generates a 1 mm inhibition zone diameter in this assay system. The preparation used in the animal trial had a titer of 1.28 × 10² AU/mL”.
Comments 9: Line 118: "two of which was challenged" – Grammar. Change to "two of which were challenged" or, better, rephrase the entire sentence for clarity. The group description is confusing.
Response 9: Thank you for your valuable comments. Lines 141-143 have been changed as: “assigned to three treatment groups (n=15): an untreated control group, a S. pullorum-challenged group, and a group that received both PP and S. pullorum challenge.”
Comments 10: Figure 1: The schematic is helpful but not referenced in the text (no callout like "Figure 1")
Response 10: Thank you for your valuable comments. Figure 1 has been cited as “Line 143 As illustrated in Fig. 1”.
Comments 11: The timeline of infection (days 10-16) and PP treatment (starting day 16) is clear but unusual. PP was administered after the infection protocol ended.
Response 11: Thank you for your valuable comments. Before starting the experiment, we reviewed literature and consulted with animal medicine professional teacher, we aimed to specifically evaluate the therapeutic efficacy of PP against an established infection, by completely separating the treatment phase from the infection phase, we can clearly demonstrate that the observed beneficial effects, including its effects on improving gut microbiota and barrier function. This provides more direct evidence for PP's application as a targeted therapeutic agent.
Comments 12: Table 1: The primer for `CLL5` seems erroneous (Forward and Reverse sequences are identical). Is this a typo? Also, `TNF-α` and `TGF-β1` primers are listed, but their accession numbers are from different databases (MF000729.1 vs NC_052569.1). This is acceptable but should be consistent.
Response 12: Thank you for your valuable comments. We have corrected in table 1 CLL5 forward primer sequences for [F: caaccgtgtgctgcttcaactat]. And the login numbers of TNF-α and TGF-β1 have been corrected to ensure that these login numbers are derived from the NCBI database.
Comments 13: The housekeeping gene is listed as `GAPDH` in the table but referred to as `β-actin` in the text (Line 175). Major inconsistency. Use one consistently
Response 13: Thank you for your valuable comments. Lines 201 has been changed as: “GAPDH”
Comments 14: The y-axis label in Figure 2 for CREA says "Cera creatinine" – Typo. Should be "Serum creatinine
Response 14: Thank you for your valuable comments. The y-axis label in Figure 2 has been changed as: “Serum creatinine”
Comments 15: Line 209: "CERA" is used in the text, but the figure label and legend use "CREA". Inconsistency. Standardize to "CREA" or "Cr".
Response 15: Thank you for your valuable comments. Lines 233 has been changed as: “CREA”
Comments 16: Line 235: "Figure 5 and table 3" – Error. It should be "Table 2" (found at the end of the manuscript). Table 2 is misplaced; it should be in the results section after Figure 5.
Response 16: Thank you for your valuable comments. Lines 246 and 254 has been changed as: “Table 2”. Table 2 has been moved the “Results” section.
Comments 17: Figure 5: Images are small. Higher magnification insets would help visualize villus/crypt differences.
Response 17: Thank you for your valuable comments. We have revised Figure 5 for clearer observation.
Comments 18: Figure 7A Label Error: The legend says "SPA" but the figure bar is labeled "SPA" for the orange bar. The text in the figure says "CON, SPA, SPA". This is incorrect. It should be CON, SP, SPA
Response 18: Thank you for your valuable comments. We have modified the SPA to SP in Figure 7A.
Comments 19: The results are clearly described, but the narrative would benefit from stating the direction of change (up/down) more explicitly.
Response 19: Thank you for your valuable comments. We carefully read the "Results" section and revised the descriptions of Line 264-267, Line 270-273 and Line 277-280.
Line 264-267: At 12 days post intervention, the expression levels of the TJ genes in the jejunal mucosa increased significantly in the SPA group and approached the values for the control group compared with those noted for the SP group.
Line270-273: At 12 DPI, the expression of chemical barrier–related genes MUC2 in the jejunum in the SPA group significantly increased, while the expression of TFF3 significantly decreased and approached the expression noted in the control group when compared with that in the SP group.
Line277-280: After 6 days of intervention, the expressions of jejunal immune-related genes TLR2, TNF-a, IL-1b and IL-6 in the treatment group decreased significantly, while TGF-b1 increased significantly and approached those of the control group when compared with that in the SP group.
Comments 20: Lines 302-303: Text says Proteobacteria and Verrucomicrobia were "significantly reduced" at 6 DPI in SP and SPA vs. CON. However, Figure 9B does not visually support a major reduction for these phyla. The statistical analysis must be shown (e.g., in a supplementary table). This is a potential inconsistency between the text and the figure.
Response 20: Thank you for your valuable comments. We have verified our statistical analysis. At 6 DPI, the reduction in Proteobacteria and Verrucomicrobia at SP/SPA compared to CON was indeed statistically significant (P < 0.01). To provide full transparency, we have added a new Supplementary Table S2 (Line 324) with complete post-hoc statistics for the microbial taxa mentioned, including p-values for these specific comparisons.
Comments 21: Line 305: "PP supplementation significantly increased the relative abundances of Firmicutes and Actinobacteriota" – Check Figure 9E; the increase in Actinobacteriota is visible, but Firmicutes appear similar between SP and SPA. Again, statistics are needed.
Response 21: Thank you for your valuable comments. We have re-examined the statistical analysis for Firmicutes and Actinobacteriota. The increases mentioned are supported by significant p-values. These detailed statistical results are now included in Supplementary Table S2 (Line 324)
Comments 22: The term "Micrococcaceae" (Line 460) is a family, not a phylum. The intended phylum is likely Actinobacteriota.
Response 22: Thank you for your valuable comments. Line 464 has been deleted: “Micrococcaceae”
Comments 23: Line 320: "UREA levels were lower in the SP and SPA groups..." This contradicts Figure 2, which shows UREA levels are higher in SP and SPA at 6 DPI.
Response 23: Thank you for your valuable comments. Lines 326-330 have been changed as: “A previous study has reported that Salmonella infection significantly increases serum alanine aminotransferase, aspartate aminotransferase, and alkaline phosphatase activity, as well as UREA and CREA level[28]. Our findings revealed that S. pullorum infection raised the levels of UREA and CREA, while its level was close to the control group when supplemented with pediocin PP”
Comments 24: Lines 359-360: "Pathogen invasion activates the innate immune system of the host" – Grammar. "of the host" -> "in the host"
Response 24: Thank you for your valuable comments. Line 362 has been changed as: “Pathogen invasion activates the innate immune system in the host”
1.The dose of PP is not quantified in bioactive units.PP
Response: We thank the reviewer for this critical comment. In our laboratory, the crude PP preparation was determined by agar well diffusion assay and is expressed as Arbitrary Units per mL (AU/mL), 1 AU is defined as the amount producing a 1 mm inhibition zone under our assay conditions. The preparation used in all experiments had a titer of 1.28 × 10² AU/mL. therefore, in the Materials and Methods section (2.2), Line 137-142 has been changed as: “The antimicrobial titer of the pediocin PP preparation was determined against Salmonella spp. using the agar well diffusion assay with an indicator lawn of approximately 3×10⁷ CFU per plate. The activity was calculated as Arbitrary Units per milliliter (AU/mL), where 1 AU is defined as the amount of bacteriocin that produces a 1 mm inhibition zone diameter under the aforementioned conditions. The preparation used in the animal trial had a titer of 1.28 × 10² AU/mL”.
- PP treatment began after infection, which may not reflect real-world prophylactic use.PP
Response: Thank you for your valuable comments. Before starting the experiment, we reviewed literature and consulted with veterinary, we aimed to specifically evaluate the therapeutic efficacy of PP against an established infection, by completely separating the treatment phase from the infection phase, we can clearly demonstrate that the observed beneficial effects, including its effects on improving gut microbiota and barrier function. This provides more direct evidence for PP's application as a targeted therapeutic agent. However, the possibility of using PP as a routine preventive additive still needs to be further evaluated through future experiment.
- The use of SPF chickens may not fully represent commercial flocks with complex microbiota.SPF
Response: Thank you for your valuable comments. The aim of this study was to evaluate the regulatory mechanism of pediocin on intestinal barrier function, renal injury, and immune regulation were evaluated in Salmonella pullorum infected chickens. Before starting the experiment, we reviewed literature and consulted with animal medicine professional teacher, the main reason is that SPF chickens can minimize the influence of other factors to the greatest extent. However, we are conducting a large-scale chicken breeding experiment, using preventive measures rather than establishing a model of Salmonella infection.
- The mechanisms linking microbiota changes to improved barrier function are correlative, not causative, in this study.
Response: Thank you for your valuable comments. We analyzed the effects of pediocin on intestinal barrier function and microbiota composition, and demonstrated its regulatory mechanism, it will provide a scientific basis for the development of new feed additives. We think that there is a two-way regulatory and mutually dependent core relationship between the intestinal microbiota and the improvement of intestinal barrier function.
- General Issues & English Corrections
Response: Thank you for your valuable comments. We have carefully proofread the entire manuscript to correct grammatical errors, improve sentence clarity, and ensure consistency in terminology and formatting.
Once again, thank you very much for your helpful comments and suggestion. We have revised the manuscript accordingly. The changes do not affect the content or framework of the paper. We earnestly appreciate your work, and we hope that the corrections will meet with your approval.
Thank you and best regards.
Yours sincerely,
Chenxin Zhou
Dongyan Zhang
zhdy203@126.com
Institute of Animal Husbandry and Veterinary Medicine, Beijing Academy of Agriculture and Forestry Sciences, Beijing 100097
Reviewer 2 Report
Comments and Suggestions for Authors
Please see the attached.

Author Response

(The authors gave the same response as above.)

Reviewer 3 Report
Comments and Suggestions for Authors
Title: Dietary Pediocin supplementation restores intestinal barrier function and microbiota balance in Salmonella-infected specific-pathogen-free chickens
This manuscript is well-written by the authors. I do believe that they can improve the manuscripts following all comments. It might have a chance to publish in the journal.
Comments
- Line 16, 95, and the whole manuscript: Please write the scientific name in italic.
- Line 17: Please re-write the following words, “Forty-five 7-day-old specific-pathogen-free (SPF) chickens”.
- Line 76-78: Please modify.
- Line 118-119: Please modify.
- Figure 1: Please modify caption. Please add the details of this figure.
- Line 176: Please recheck the format of formulation (2-ΔΔCt).
- Table 1: Please check the writing of target gene names. It should be written in italic.
- Please check the term UREA in figure 2 to match your methods section.
- Line 234: Please check the information.
- Figure 9: Please increase the magnification.
- Discussion: Try to compare the results (the author’s hypothesis) with other finding by other researchers. Please delete some sentences like introduction and result from discussion.
- Please delete old references. The references of the years 2020-2025 should be added.
Author Response
Dear Reviewer,
Thank you for your letter and the reviewers’ constructive comments on our manuscript “Dietary Pediocin supplementation restores intestinal barrier function and microbiota balance in Salmonella-infected specific-pathogen-free chickens” (ID: ISSN 2076-2607). We have carefully considered all suggestions and have revised the manuscript accordingly. Our point-by-point responses are provided below:
Comments 1: Line 16, 95, and the whole manuscript: Please write the scientific name in italic.
Response 1: Thank you for your valuable comments. We have gone through the entire manuscript and corrected the formatting of all scientific names (genus and species) to be in italic, as per standard convention.
Comments 2: Line 17: Please re-write the following words, “Forty-five 7-day-old specific-pathogen-free (SPF) chickens
Response 2: Thank you for your valuable comments. Lines 16-17 have been changed as: “Forty-five 7-day-old specific pathogen free (SPF) chickens”
Comments 3: Line 76-78: Please modify
Response 3: Thank you for your valuable comments. We have improved the wording on Lines 76-78 to make it clearer and more accurate.
Comments 4: Line 118-119: Please modify.
Response 4: Thank you for your valuable comments. We have improved the wording on lines 118-119 to make it clearer and more accurate.
Comments 5: Figure 1: Please modify caption. Please add the details of this figure
Response 5: Thank you for your valuable comments. We have revised the caption of Figure 1 to include a more detailed description of the experimental groups, treatments, timelines, and key procedures as recommended. For example, figure 1. Schematic overview of the experimental design. Forty-five 7-day-old specific pathogen free (SPF) chickens were randomly assigned to three groups: control (CON), Salmonella Pullorum infection (SP), and infection plus pediocin PP treatment (SPA). Starting on Day 3 (10 days of age), chickens in the SP and SPA groups were orally inoculated with S. pullorum (1.0 × 10⁷ CFU/mL/day) for 6 consecutive days, while CON group received sterile LB medium. From Day 9 (16 days of age), chickens in the SPA group received daily oral administration of pediocin PP preparation (1.28 × 10² AU/day) for 6 days, while the CON and SP groups received sterile YPD medium. Key sampling time points are indicated. Arrows indicate the timing of interventions. (Line 484-492)
Comments 6: Line 176: Please recheck the format of formulation (2-ΔΔCt)
Response 6: Thank you for your valuable comments. Lines 202 has been changed as: “2-ΔΔCt”
Comments 7: Table 1: Please check the writing of target gene names. It should be written in italic
Response 7: Thank you for your valuable comments. We have checked and reformatted all target gene names in Table 1 to be written in italic as per the nomenclature rules for avian genes.
Comments 8: Please check the term UREA in figure 2 to match your methods section.
Response 8: Thank you for your valuable comments. Lines 497 has been changed as: “UREA: Serum urea”
Comments 9: Line 234: Please check the information.
Response 9: Thank you for your valuable comments. We have improved the wording on line 234 to make it clearer and more accurate.
Comments 10: Figure 9: Please increase the magnification
Response 10: Thank you for your valuable comments. The magnification in Figure 9 has been increased as recommended to show finer details.
Comments 11: Discussion: Try to compare the results (the author’s hypothesis) with other finding by other researchers. Please delete some sentences like introduction and result from discussion.
Response 11: Thank you for your valuable comments. We have revised the Discussion section, and the manuscript has been undergone language editing.
Comments 12: Please delete old references. The references of the years 2020-2025 should be added.
Response 12: We thank the reviewers for their suggestions to update the literature. We have systematically reviewed the reference list and have removed outdated references.
Once again, thank you very much for your helpful comments and suggestion. We have revised the manuscript accordingly. The changes do not affect the content or framework of the paper. We earnestly appreciate your work, and we hope that the corrections will meet with your approval.
Thank you and best regards.
Yours sincerely,
Chenxin Zhou
Dongyan Zhang
zhdy203@126.com
Institute of Animal Husbandry and Veterinary Medicine, Beijing Academy of Agriculture and Forestry Sciences, Beijing 100097

Round 2
Reviewer 1 Report
Comments and Suggestions for Authors
Well revised.